# Multi-Analytical Approach to Characterize the Degradation of Different Types of Microplastics: Identification and Quantification of Released Organic Compounds

**DOI:** 10.3390/molecules28031382

**Published:** 2023-02-01

**Authors:** Giulia Giaganini, Mario Cifelli, Denise Biagini, Silvia Ghimenti, Andrea Corti, Valter Castelvetro, Valentina Domenici, Tommaso Lomonaco

**Affiliations:** Department of Chemistry and Industrial Chemistry, University of Pisa, 56124 Pisa, Italy

**Keywords:** microplastics, nanoplastics, high resolution nuclear magnetic resonance, ^1^H NMR, GC-MS, artificial aging, PET, PP, PS, HDPE, LDPE

## Abstract

Microplastics and nanoplastics represent one of the major environmental issues nowadays due to their ubiquitous presence on Earth, and their high potential danger for living systems, ecosystems, and human life. The formation of both microplastics and nanoplastics strongly depends on both the type of pristine materials and the degradation processes related to biological and/or abiotic conditions. The aim of this study is to investigate the effect of two of the most relevant abiotic parameters, namely temperature and light, taken under direct control by using a Solar box, on five types of reference polymers: high density polyethylene (HDPE), low density polyethylene (LDPE), polypropylene (PP), polystyrene (PS), and polyethylene terephthalate (PET). A multi-analytical approach was adopted to investigate in detail the first steps of plastics degradation. Samples of plastic materials at different degradation times were analyzed by means of ^1^H NMR spectroscopy and thermal desorption gas chromatography mass spectrometry (TD-GC-MS) technique. Several minor molecular species released during degradation were consistently identified by both techniques thus providing a comprehensive view of the various degradation products of these five types of microplastics.

## 1. Introduction

In recent years, microplastics (MPs) and nanoplastics (NPs) are being recognized as one of the main environmental problems due to their ubiquitous presence in terrestrial and aquatic ecosystems. However, to date, the actual concentration levels of both MPs and NPs and their effects on human health are still largely unknown [1]. Recent reports of the presence of MPs and NPs in the placenta [2] and bloodstream [3], if confirmed, suggest their potential toxicity and health risks for human health. It is widely accepted that plastic materials in the range of 1 μm to 5 mm are referred to as MPs, a classification typically based on the capture and detection thresholds of the adopted devices and techniques, whereas NPs have a dimension in the sub-micrometer range [4].

The reliable identification and quantification of MPs in environmental samples is a crucial step to devise a global strategy for an effective mitigation of the environmental and health impacts of these pollutants [5] as pointed out for other contaminants [6,7]. From an analytical point of view, Fourier transform infrared (FT-IR) and Raman spectroscopies are generally used to chemically identify MPs [5], along with micro-Raman and micro-FT-IR for the analysis of smaller particles [8]. Pyrolysis-gas chromatography/mass spectrometry (Py-GC/MS) technique is widely employed to analyze MPs due to unique advantages that are related to the need of limited sample pre-treatment as well as high sensitivity and reproducibility levels [9]. However, the presence of interfering compounds due to biogenic material of the sample and persistent organic pollutants captured from the environment could result in misleading interpretations of the analytical data [10]. In any case, the above methodologies may not be able to provide accurate data concerning both the chemical characterization of MPs and their concentration in complex samples.

A multi-analytical approach is thus necessary to achieve a comprehensive chemical characterization of environmental and biological samples, aimed at providing information essential to evaluate the extent, distribution, and environmental impact of MPs, including their progressive degradation eventually leading to nanometric particles and low molecular weight chemical compounds. In this regard, a successful procedure allowing a polymer-specific exhaustive and accurate analysis of the total mass MPs and NPs contaminants in marine and freshwater sediments has been reported [11]. This so-called PISA (Polymer Identification and Specific Analysis) procedure, based on selective solvent extraction/fractionation (for polyolefins, polystyrene, and other vinyl polymers) followed by hydrolytic depolymerization (for condensation polymers such as polyethylene terephthalate, PET, and the two polyamides nylon 6 and nylon 6,6), along with quantification by chromatographic techniques (Py-GC/MS and size exclusion chromatography, SEC, for extractable polymers; reversed phase HPLC for depolymerization products) [12], is being further expanded into a multi-analytical platform including a range of mass spectrometer-based methods [13] and techniques (selected-ion flow tube mass spectrometry) [14], aimed not only at determining MPs and NPs in environmental matrices, but also at investigating their degradation by analyzing the chemical products released as volatile organic compounds (VOCs), especially as a result of MPs exposure to environmental photo-oxidative conditions.

High resolution ^1^H NMR spectroscopy of solutions and of liquid complex mixtures is a well-known analytical technique mainly used to identify chemical compounds and to elucidate the chemical structure of both molecules and macromolecules in solution. In the last three decades, quantitative NMR (qNMR) spectroscopic methods allowing the identification and quantification of chemical compounds even at low concentration in complex mixtures have been developed [15], with a significant growth of the scientific research based on qNMR in such fields as metabolomics, pharmaceutics, nutraceutics, and chemical–physical analysis of natural products [16,17]. The most common qNMR techniques are based on the detection of protons (^1^H) in chemical compounds, while few qNMR studies are based on ^13^C and ^31^P nuclei, due to their lower relative natural abundance [15] resulting in lower sensitivity. Quite recently, qNMR has been proposed for purity analysis as an alternative to the mass balance method [15,18,19]. Moreover, several techniques based on internal, external, or electronic referencing can be adopted reaching high analytical standards [20]. Despite the higher limit of detection (LOD) and limit of quantification (LOQ) with respect to other analytical techniques, the non-destructive character, the possibility to detect at the same time all minor components without need of separating different classes of chemical compounds, and the increased accuracy and precision of the most recent NMR instruments qualify NMR as a powerful analytical technique [15,20].

Recently, ^1^H NMR spectroscopic methods as well as qNMR techniques have been also applied to the study of microplastics made of polyethylene (PE), polyethylene terephthalate (PET), polystyrene (PS) [21], polyvinyl chloride (PVC), acrylonitrile butadiene styrene (ABS), and polyamide (PA) [22]. These studies demonstrated that it is possible to identify and quantify each microplastic type thanks to the presence of ^1^H NMR signals specific to the pristine plastic material. Moreover, Peez et al. [23] showed that in the case of PET microplastics derived from different environments and subjected to various external conditions, qNMR methodologies can achieve very high analytical standards when associated with appropriate calibration and sampling procedures. It is worth noticing that other NMR techniques, such as 2D ^1^H NMR diffusion-ordered spectroscopy (DOSY) [24], low field ^1^H NMR [25], and ^13^C multi cross polarization (CP) [26], were recently used to investigate microplastics.

Multi-analytical approaches which include NMR spectroscopic techniques [24,27] are thus expected to provide a more solid understanding of the chemical nature of microplastics, a better chemical identification of different microplastic types [21,28] and, as presented in this work, to identify minor compounds released from microplastics during degradation [29,30].

Accordingly, in this paper, we are extending our analytical platform, including ^1^H NMR spectroscopy to further characterize the degradation products released from different types of microplastics artificially aged inside a Solar box, with controlled temperature, humidity, and light irradiation, for a period of 1 month. Microplastics used as reference samples were made of low-density polyethylene (LDPE), high density polyethylene (HDPE), polyethylene (PE), polyethylene terephthalate (PET), polypropylene (PP), and polystyrene (PS) with average dimensions from 509 μm to 857 μm. The degradation products of these five types of microplastics were monitored through ^1^H NMR spectroscopy and an optimized HiSorb extraction coupled to thermal desorption gas chromatography mass spectrometry (HiSorb-TD-GC-MS) technique for the analysis of the head space (HS). The results obtained show that the two techniques are consistent and that the information about the minor chemical compounds produced during degradation, as obtained with this multi-analytical approach, could be used as a basis for future investigations of more complex environmental samples.

## 2. Results

### 2.1. Qualitative and Quantitative Characterization of Polymer Materials with HiSorb-TD-GC-MS Analysis

Figure 1 shows the total ion chromatograms (TIC) acquired at t_0_ and after 4 weeks (4 w) of irradiation from each investigated polymer.

The head space fraction of the five aged samples (t_0_ + 4 w) from the different polymers, analyzed after adsorption/desorption through HiSorb coupled to TD-GC-MS, highlighted complex chromatograms with clear cases of co-elution. The deconvolution process allowed us to identify hundreds of compounds (only about 10% of the peaks) mainly belonging to ketones, aldehydes, carboxylic acids, and aromatic compounds. For all aged materials, most of the GC signals were in the second part of the chromatograms (from 30 min to 60 min) where C5–C18 compounds are generally eluted. Similar results had been observed in a previous study where Tenax-GR-based needle trap device technique has been employed to characterize the VOCs profile released from aged MPs [13]. The compounds identified from the different virgin polymers and the corresponding (t_0_ + 4 w) samples are listed in Appendix A.

Figure 2 shows the results of the principal component analysis (PCA) concerning the HiSorb-TD-GC-MS data.

In PCA, the first principal component (PC1) and the second principal component (PC2) describe 38% of the total variance, confirming the complexity of the investigated system. From the score plot, it is possible to distinguish three clusters: (i) a first group of points in the bottom-left, (ii) a second group in the center-right, and (iii) a third group in the center. Interestingly, the first and second group belonged to PS and PP samples, whereas the third group combined the three remaining polymers (i.e., PET, HDPE, and LDPE). The loading plot allowed us to find the most important variables characterizing the PS and PP groups, suggesting the possibility to distinguish different aged polymers according to their VOCs profile. In fact, HiSorb-TD-GC-MS data suggested that degraded PS released significant amounts of the aromatic compounds acetophenone, benzaldehyde, toluene, and benzene, whereas aged PP showed acetic acid, acetone, 2-butanone, and 2-pentanone. Moreover, the occurrence of HDPE, LDPE, and PET in a single cluster close to the center of the score plot suggests a similar VOCs profile for these polymers.

### 2.2. H NMR Spectroscopy of Polymer Microplastics during Degradation

In this section, the main features of the ^1^H NMR spectra for the five types of polymer materials from the virgin sample to samples under different times of artificial photo-aging will be presented, following the order PS, PP, PET, HDPE, and LDPE.

Figure 3 shows the ^1^H NMR spectra of the photo-aged PS samples registered in solution.

Our results showed that PS can be completely dissolved in CDCl_3_, irrespective of the degradation level, by shaking the solution at 80 °C for about 30′, confirming data reported elsewhere [21]. The ^1^H NMR spectrum of virgin PS microplastics is characterized by two groups of structured signals between 6.3 and 7.2 ppm ascribable to the aromatic protons (^2^H in ortho, ^2^H in meta, and ^1^H in para positions) of the styrene moiety (see Figure 3a) and a group of broad signals centered at 1.3 and 1.8 ppm due to the CH and CH_2_ protons of the polymer chain, respectively. A sharper signal at 0.8 ppm is due to the CH_3_ terminal groups of PS, as also reported elsewhere [31]. An enlargement of the baseline put in evidence the presence of very small signals between 2.3 ppm and about 5 ppm, which can be due to oxidized groups [31] already present in the pristine PS microplastics (see Figure 3b,c). On the contrary, the spectral region above 7.5 ppm of the pristine PS microplastics does not show any signals (see Figure 3d). The effect of degradation under artificially simulated environmental conditions can be observed by comparing the enlargement of the ^1^H NMR spectra close to the spectral baseline at different times of irradiation (see Figure 3b–d). As it will be discussed in Section 3, some spectral changes are observed in the aliphatic region between 2.30 and 2.65 ppm (see Figure 3b); however, the most significant spectral changes are observed in the region between 7.0 and 8.5 ppm, with the appearance of structured and well-defined signals whose relative integrals increase with increasing irradiation time. These spectral variations indicate the presence of specific degradation products, partially in agreement with previous studies on environmental degradation of PS [32].

Figure 4 shows the ^1^H NMR spectra of the photo-aged PP samples.

The dissolution of PP in organic solvents, and thus the acquisition of ^1^H NMR spectra of polypropylene in solution, requires higher temperatures [33]. In our study, the acquisition of the ^1^H NMR spectra of the pristine PP (t_0_) microplastics had to be performed at 90 °C. The PP samples at different times of irradiation were more easily solubilized than PP (t_0_). This aspect could be related to a microstructure-dependent sensitivity to photo-oxidation. As it is shown in Figure 4a, the ^1^H NMR spectrum of PP (t_0_) contains signals typical of both isotactic and syndiotactic PP [33]. The triplet centered at 1.15 ppm is a unique characteristic of syndiotactic PP, while the signal from the CH_3_- group in syndiotactic PP is shifted to lower chemical shift than in isotactic PP [30]. As seen in Figure 4a, the ^1^H NMR spectrum of artificially aged PP (from t_0_ + 1 w to t_0_ + 4 w) contains only signals due to isotactic PP. While all degradation processes typically take place preferentially in the amorphous phase, the observed change in tacticity is in agreement with the results reported in a recent paper [34], suggesting a preferential degradation of syndiotactic PP. Another important feature concerns the appearance of several structured signals in the region between 3 ppm and 6 ppm in the spectra of PP upon aging (see Figure 4b), which are totally absent in the pristine PP (t_0_). As it will be discussed in Section 3, the appearance of new signals between 3 and 6 ppm, and above 7.5 ppm, as well as the changes in their relative intensity during the progress of degradation (see proton signals from ‘a’ to ‘e’ in Figure 4b) can be associated to the presence of specific degradation products or structures [35,36,37].

Figure 5 shows the solution ^1^H NMR spectra of the photo-aged PET samples.

PET particles, in the form of fibers, microplastics, and films, have been previously studied by means of ^1^H NMR spectroscopy [21,23,31,38]. In our study PET microplastics were solubilized in a CDCl_3_/TFA (3/1 *V*/*V*) mixture [21], and the ^1^H NMR spectra were acquired after 30′ at 80 °C. The observed characteristic signals of PET are an intense singlet at 8.11 ppm due to the aromatic protons (^4^H in the phenyl ring), an intense signal at 4.74 ppm, and a less intense signal at 4.1 ppm ascribable to the CH_2_ close to the ester and ether functional groups, respectively [38]. As shown in Figure 5, all photo-aged PET microplastic samples present additional small signals between 0 and 4 ppm, which can be due to the presence of CH and CH_2_ signals close to ketones, aldehydes, and carboxyl functional groups, as it will be discussed in Section 3. Among these proton signals, some show either an increase or a decrease in relative intensity, thus indicating changes occurring to the PET macromolecular structure as a result of photo-induced degradation [27,29,37].

Figure 6 shows the ^1^H NMR spectra of the photo-aged HDPE samples registered in solution.

Similar to PP, the dissolution of HDPE in organic solvents can only be achieved at high temperature. HDPE microparticles were treated with toluene-d_8_ at about 80 °C for more than 30′ for a complete solubilization. ^1^H NMR spectra of HDPE are characterized by two groups of signals: one intense signal is centered at 1.33 ppm (due to the CH_2_ in the chain) and one small signal is centered at 0.9 ppm (due to the terminal CH_3_) [39]. Signals at 2.1 ppm and at 6.95–7.10 ppm are due to the solvent (toluene-d_8_). In the pristine HDPE (t_0_), no additional signals are observed in the ^1^H NMR spectrum, while in the four samples analyzed after different irradiation times, several signals in the region between 3 and 6 ppm appear. As shown in Figure 6, almost all new signals (see blue circles) increase their relative intensity with increasing irradiation time.

Figure 7 shows the ^1^H NMR spectra of the photo-aged LDPE samples registered in solution.

The ^1^H NMR spectral features of LDPE are similar to those of HDPE, with the exception that the relative intensity between the signals at 0.9 ppm (CH_3_) and 1.33 ppm (CH_2_) is different in the two polymers due to the presence of chain branching in LDPE [39]. In the pristine LDPE (t_0_), very small signals in the 4.7–5.1 ppm and 5.6–5.7 ppm regions indicate the presence of olefines (see Figure 7), as discussed in Section 3. The spectra of the LDPE samples analyzed after different irradiation times (from t_0_ + 1 w to t_0_ + 4 w) show additional ^1^H NMR signals in the 2.3–2.7 ppm and 7.4–7.8 ppm regions, which increase their intensity during degradation.

## 3. Discussion

PDMS-HiSorb probes is a reliable absorptive extraction tool normally used to extract VOCs and SVOCs by its direct immersion in any liquid media [40]. In this work, we tested for the first time the potential advantages of this technique to analyze the HS fraction of MPs incubated for 3 h at 60 °C. The technique allowed us to extract a great number of compounds emitted from MPs during the degradation processes. Compared to the results obtained with the needle trap microextraction (NTME) technique, previously used in our lab for the analysis of the same set of aged samples [13], we observed a significant increase in the number of detected signals, highlighting the capability of HiSorb to promote the extraction of VOCs from solid samples. This is probably due to the larger sorbent phase volume of the HiSorb with respect to that packed within the needle trap device. Preliminary qualitative chemical characterization of VOCs profile released during the degradation was carried out using the Unknown™ analysis tool (Agilent Technologies, Santa Clara, CA, USA) that revealed the presence of about 1200 signals in the chromatograms of PS, PP, HDPE, and LDPE, whereas PET showed only 250 signals, probably due its greater stability to photo-oxidation processes [41]. Due to the complexity of the chromatogram, only a small portion (about 10%) of these signals was successfully identified by GC-MS analysis. Most of the compounds belonged to ketones, aldehydes, carboxylic acids, lactones, esters, and aromatic compounds, in agreement with the results previously obtained with NTME technique [13]. PCA suggested the potential use of HS analysis to distinguish different polymers based on their VOCs profile, and to assess the associated toxicity potential. As an example, aged PS released mainly aromatic compounds such as acetophenone, benzaldehyde, toluene, and benzene, all of them with well-known toxicity.

In Section 4 of the paper, we are presenting the identification of the most significant signals in the ^1^H NMR spectra of the five polymer types and their possible evolution with the progress of degradation at increasing irradiation times. The identification of almost all signals was performed by comparing the experimental spectra obtained in this study with the experimental spectra of minor compounds available in open access spectra databases and in the scientific literature, and ^1^H NMR spectra predicted by using the simulation tool, as reported in Section 4.5.

^1^H NMR spectra of pristine PS (t_0_) and photo-aged PS (from t_0_ + 1 w to t_0_ + 4 w) are reported in Figure 3. Generally, we observed narrowed NMR signals that exclude the presence of functional groups in the macromolecular structure of the polymers. Among the minor degradation compounds identified by HiSorb-GC-MS analysis (Appendix A), specific ^1^H NMR signals were identified for the following compounds: methyl salicylate, acetic acid, phenyl methyl ester, cinnamaldehyde (E), acetic acid phenyl ester, benzyl alcohol, and 2-hydroxy-benzaldehyde. Some of these compounds are shown in Figure 8 and some of the typical proton signals are underlined. All aromatic protons of the bicyclic benzofuran could be assigned to experimentally observed signals. Similarly, the aromatic protons benzyl alcohol and methyl salicylate are put in evidence in Figure 8 in the regions 7.30–7.45 ppm and centered at 7.55 and 7.96 ppm. In these spectral regions, several other derivatives of salicylates could be present [42]. The intensity of the aromatic signals centered at 7.47, 7.48, and 8.08 ppm increase because of degradation. None of the compounds identified by GC-MS present proton resonances at these frequencies; however, PS degradation is expected to release several additional compounds such as styrene derivatives, poly condensate aromatic systems, and benzophenone derivatives such as the chalcones [43] reported in Figure 8. The presence of compounds such as 2-hydroxy-benzaldehyde is also supported by the presence of signals in the region between 9.95 and 10.05 ppm (not shown here).

Focusing on the most intense signals among the products of degradation of PS, underlined in Figure 8 with the blue, green, and yellow braces, the ratio R = I_DP_/I_PS_ between the intensity of the signals of degradation products and the intensity of the aromatic signals of PS (5H) was found to increase with increasing degradation time for all signals.

The degradation of PP under the experimental conditions described in Section 4 produced a relatively large number of low molecular weight compounds, as identified by HiSorb-TD-GC-MS analysis (see Appendix A). ^1^H NMR spectra of PP at different times of degradation are rich with minor signals in the region between 2.7 and 6.1 ppm, as shown in Figure 4 and enlarged in Figure 9. Several ^1^H NMR signals are compatible with low molecular weight molecules identified by HiSorb-TD-GC-MS (Appendix A), such as (R)(-)2-pentanol, acetylacetone, 1.1-ethanediol diacetate, 2-butenoic acid (E), 1-heptanol, pentanoic acid, and 1-(acetyloxy)-2-propanone. While the presence of other compounds, such as 2-pentanone and other ketones, cannot be excluded from ^1^H NMR spectra, their characteristic signals would be partially overlapped with those of pristine PP.

In the ^1^H NMR spectra of the irradiated PP samples (from t_0_ + 1 w to t_0_ + 4 w), several well-defined signals could be assigned based on the analysis of their multiplicity. For instance, the signal at 5.93 can be assigned to the CH- in position 2 of the 2-butenoic acid (E), while signals between 4.95 ppm and 5.80 ppm are due to 1-alkene and 2-alkene as the ones shown in Figure 9.

^1^H NMR signals due to specific products of degradation of PP are the singlets in the region between 4.65 and 4.85 ppm. As shown in Figure 9, some of these signals are ascribable to molecular species with CH- linked to the alcoholic function in 2-methyl 3-propanol derivatives, such as in isopropyl ethers or esters. The singlet at 4.68 ppm is due to CH_2_- of the 2-butanone acetate. Less intense and partially overlapped signals appear after 1 week of degradation in the region between 3.1 ppm and 4.2 ppm. Here, several characteristic compounds can be identified among typical products of PP degradation. For instance, the protons of CH_2_- linked to -OH in 3-methyl butanol resonate at 3.16 ppm (triplet) and those of the CH_2_- in acetyl acetone resonate at 3.56 ppm. If we look at the ratio between the intensity of these degradation products and the intensity of CH_3_- in pristine PP material, we cannot see significant changes between 1 w and 4 w, except for a few signals.

The analysis of the ^1^H NMR spectra of photo-oxidized PET is more complicated with respect to the previous polymers due to the presence of several signals in the region between 1 and 4 ppm and between 7.3 and 8.8 ppm which are present in all samples, from the pristine PET (t_0_) to the PET (t_0_ + 1 w, t_0_ + 2 w, t_0_ + 3 w, and t_0_ + 4 w) samples (see Figure 5). Moreover, some signals do not change their intensity, such as the singlets at 2.52 ppm and at 2.31 ppm, which can be ascribed to the -CH_3_ of acetophenone and 2-ketones. Some proton signals decrease their intensity during degradation, such as the triplet centered at 0.87 ppm which is due to terminal CH_3_ in ethyl groups which can be associated with many low molecular weight species released during PET degradation. As observed also from the HiSorb-TD-GC-MS results, in the case of PET, the small number of identified molecules is also reflected in the relatively simpler ^1^H NMR spectra, tentatively assigned to compounds such as nonanal, octanal, butanal, and butanone derivatives, as well as several benzene derivatives as detected by GC-MS.

The ^1^H NMR spectra of HDPE in toluene-d_8_ (shown in Figure 6) present significant changes because of degradation in the 3–6 ppm region. Signals centered at 3.35–3.45 ppm are typical of methylene with a hydroxyl group (-CH_2_-OH). Low molecular weight alcohols were also found by HiSorb-TD-GC-MS (see 1-decanol in Appendix A). The well-defined doublet (partially overlapped) signals between 4.90 and 4.99 ppm are most probably due to not equivalent CH_2_- ending group of alkene compounds, such as 1-pentene, and this is also in agreement with the presence of another signal (more structured) centered at 5.76 ppm, which is ascribable to the -CH- of the ethenyl group. Low molecular weight oxidized alkene derivatives such as methyl vinyl ketone were also found by HiSorb-TD-GC-MS (Appendix A). A significant ^1^H NMR signal appearing after 2 weeks of degradation, centered at 5.45 ppm, is typical of the -CH (ethenyl group) linked to the -CH_3_ group as in the trans-butene or 2-cis-pentene compounds. A more quantitative spectral analysis can be performed by comparing the NMR integrals ascribable to the polymer chain, namely the signal centered at 1.33 ppm due to CH_2_- of the HDPE chains (I_HDPE_) and those of the product of degradation (I_DP_). As it can be seen in Figure 10, the ratio R = I_DP_/I _HDPE_ increases during degradation time (from t_0_ to t_0_ + 4 w) for all signals identified and described previously. After 4 weeks of polymer degradation, the ratio R is about 0.00017 for the signal ascribable to alcohols and it reaches the value of 0.00087 for the signal due to terminal CH_2_- of 1-alkene (see, for instance, some representative chemical compounds in Figure 10). Knowing the amount of the pristine polymer (at t_0_), we roughly estimated the concentration of minor compounds in the range of 1–10 μg/mL, which is coherent with the NMR sensitivity, highlighting the potential role of ^1^H NMR to analyze aged polymers aimed at monitoring degradation products.

As reported in Section 2 (Figure 7), the ^1^H NMR spectra of LDPE (pristine sample and samples at different times of degradation) are quite similar to those recorded on HDPE. From the point of view of the degradation products, LDPE shows typical signals of terminal 1-alkene, as well as 2-alkene compounds, similarly to HDPE microplastics under degradation. As observed in Figure 11, after 4 weeks of degradation, the ^1^H NMR signals centered at 4.90–4.99 ppm (dominated by two doublets) and at 5.74–5.75 ppm (a multiplet) have relatively high intensities with respect to other minor signals. Moreover, they are well-defined and structured signals, in agreement with ^1^H NMR spectral signals of 1-pentene (and other 1-alkene compounds), as reported in the open-access NMR database (see Section 4.5). The proton signal centered at 3.4 ppm is due to methylene close to a hydroxyl group (-CH_2_-OH), typical of alcohols, as observed for HDPE. Moreover, additional signals can be identified whose intensity increases by increasing the degradation time. Small multiplet ^1^H NMR signals appear in the aromatic region, between 7.55 and 7.85 ppm, and in the aliphatic region, between 2.35 and 2.65 ppm (see Figure 7). These signals could be ascribed to benzene and toluene derivatives, such as benzoic acid, p-toluic acid/o-toluic acid, and p-toluic acid. For instance, the CH_3_- group linked to the aromatic ring of toluic acids usually gives rise to signals between 2.35 and 2.65 ppm. The ^1^H signals on the aromatic ring of toluic acids are consistent with the observed signals between 7.55 and 7.85 ppm. The appearance of a signal around 3.8 ppm after 3 weeks of LDPE degradation can be due to methyl groups of toluic acid methyl ester. If we compare the results obtained from ^1^H NMR spectroscopy with those obtained by HiSorb-TD-GC-MS (see Appendix A), we can see that some of the compounds mentioned above were found by both techniques (i.e., alcohols, such as 1-nonanol; benzene derivatives, such as benzaldehyde; alkene derivatives, such as 6-heptenoic acid). Minor compounds, such as butanoic acid and 2(5H) furanone derivatives, found by HiSorb-TD-GC-MS, cannot be excluded from NMR. However, their characteristic ^1^H NMR signals are in the spectral region of the intense signals due to CH_2_- and CH_3_- groups of the pristine LDPE.

In Figure 11, the trend of the ratio R = I_DP_/I_LDPE_ is reported during the degradation. Here, the value of I _LDPE_ corresponds to the integral of the signal due to CH_2_- in the polymer chain of LDPE (see Figure 7). It can be noted that the ratio R (I_DP_/I _LDPE_) increases during degradation time (from t_0_ to t_0_ + 4 w) for all signals identified. In particular, the highest increase was observed for the two signals ascribable to 1-alkene degradation products, reaching the value of 0.031 for the ^1^H NMR signals centered at 4.90–4.99 ppm. Both values and trends of R (I_DP_/I_LDPE_) indicate that LDPE degradation is faster than HDPE, under similar experimental conditions.

The above NMR results open several possible developments. With respect to the present scientific literature [19,20,21,22,23], which was focused on the identification and quantification of different polymeric types in microplastics, either artificially prepared or found in different environments, in this work, the focus was on the chemical products of degradation. The present study demonstrates that several NMR signals can be identified and many of them can be assigned to specific chemical compounds. Moreover, as reported in the case of PS, HDPE, and LDPE materials, the trends of the intensity of NMR signals assigned to specific chemical compounds suggest that this technique is suitable to make kinetics investigations and to study the degradation mechanisms in a more quantitative way.

In conclusion, for the first time, we clearly demonstrated the potential role of ^1^H NMR to investigate the degradation products of PP, LDPE, HDPE, PS, and PET MPs under oxidative photodegradation conditions. The advantageous combination of two powerful techniques such as GC-MS and ^1^H NMR is the main novelty of this work, contributing to the improvement of the chemical characterization of such complex samples as polymers with different degradation levels, and providing useful reference data on the environmental aging of plastic debris. The determination and quantification of some of the identified compounds will be the subject of future investigations more specifically focused on the kinetics and mechanisms of degradation.

## 4. Materials and Methods

### 4.1. Chemicals and Materials

Toluene-d_8_ (vials of 0.75 ML, 99.6% deuteration), CDCl_3_ (100 g bottle, 99.8% deuteration), and trifluoroacetic acid (TFA, spectroscopic grade, 100 mL) were purchased from Merck (Sigma Aldrich, St. Louis, MO, USA). Stock liquid mixture of C5–C18 n-alkanes (pentane, hexane, heptane, octane, nonane, decane, undecane, dodecane, tridecane, pentadecane, hexadecane, heptadecane, and octadecane) was purchased from Merck (Sigma Aldrich, St. Louis, MO, USA) and used to calculate Kovatz indexes, as reported elsewhere [44].

### 4.2. Samples and Artificial Aging Procedure

Micronized powders of PP, LDPE, HDPE, PS, and PET produced by cryogenic milling were a kind gift from Poliplast S.p.A. (Bergamo, Italy), as reported in Table 1. The average particle size of each material was determined according to ISO 565–1990 regulation.

Virgin polymeric materials were artificially aged for one month, as reported elsewhere [13]. Briefly, a Solar box RH 3000 (KEWLAB, Milan, Italy), equipped with a Xenon light source and an outdoor filter, was employed to control temperature (40 °C) and power (750 W/m^2^) within the ventilated irradiation chamber to simulate the environmental exposition. An aliquot (200 mg) of each aged material was collected before (time t_0_) and after 1 (t_0_ + 1 w), 2 (t_0_ + 2 w), 3 (t_0_ + 3 w), and 4 (t_0_ + 4 w) weeks of irradiation in the Solar box, respectively, of artificial aging. The sampled powders were stored in glass vials at −20 °C until analysis.

### 4.3. Samples Preparation for the NMR Analysis

Before performing ^1^H NMR spectroscopy, tests were carried out to identify the best solvent for each polymeric material for the subsequent NMR analysis. Toluene-d8 was used to dissolve HDPE, LDPE, and PP, whereas CDCl_3_ and a 3/1 *V*/*V* mixture of CDCl_3_ and TFA was used for PS and PET, respectively. Concentrated samples were prepared by diluting the required amount of each polymer (ranging from 10 to 14 mg) in 1 mL of solvent, as reported elsewhere [21,23]. High precision NMR tubes (Wilmad^®^, Vineland, NJ, USA) loaded with the sampled polymer micro powders and the corresponding solvent were heated for 30′ up to 80 °C to completely dissolve PS, LDPE, and PET, and up to 90 °C for HDPE and PP, respectively. Tubes containing HDPE and PP samples were additionally mixed by using a LP vortex mixer (Thermo Fisher Scientifics, IT, Waltham, MA, USA) and then inserted into the NMR magnet at a temperature of 80 °C. To follow the aging process, five samples for each polymer type, from t_0_ to t_0_ + 4 w, were analyzed.

A JEOL^®^ Delta 500 spectrometer (JEOL Ltd., Tokyo, Japan) with a 500-MHz 5-mm TH ATM probe head was used for the ^1^H NMR measurements. A single pulse experiment was performed for all samples using the following acquisition parameters: receiver gain optimized for each polymer type, 90° pulse width of 5.15 μs, 32 scans, acquisition time of 1.74 s, and relaxation time of 5 s. The recorded ^1^H NMR spectra were processed and analyzed by using the software Delta 6.0 (JEOL Ltd., Tokyo, Japan). For each type of polymer, phase correction, base-line correction, and linewidth were set manually. Integrals of the characteristic signals of HDPE, LDPE, PET, PP, and PS were used to determine the relative concentration of minor chemical species released during the degradation process, as reported in Section 2 and Section 3.

### 4.4. Samples Preparation for the HiSorb-TD-GC-MS Analysis

An aliquot (10 mg) of each sample was placed into a 10 mL headspace vial sealed with a crimped HiSorb septum cap. A metal-core PDMS HiSorb was gently inserted into the vial through the septum. The vial was kept at 60 °C for 3 h for the extraction of VOCs released by each sample tested. The HiSorb probe was removed from the vial and then inserted into an empty TD tube for the analysis. A TD-100 multi-tube auto-sampler (Markes International, Cardiff, UK), equipped with an automated re-collection system, was employed for the thermal desorption of the PDMS probe at 250 °C for 15 min with a nitrogen flow rate of 50 mL/min. During the primary desorption (splitless mode), analytes were cryogenically trapped at 5 °C in an internal focusing trap packed with 70 mg of Tenax GR (Markes International, Cardiff, UK). The cold trap was then desorbed in split mode (split ratio of 11) at 300 °C for 20 min. The GC-MS analyses were performed using the same GC-MS instrumental settings reported elsewhere [45]. Briefly, an Agilent 7890 B Gas Chromatograph coupled with an Agilent 7010 GC-MS Triple Quad Mass Detector (Santa Clara, St. Louis, MO, USA) equipped with an Agilent DB-5 ms capillary column (60 m length, I.D. 0.25 mm, and 1 μm film thickness) (Santa Clara, St. Louis, MO, USA) at 1 mL/min of He as the carrier gas was employed. The oven temperature program was 30 °C for 13 min, 4 °C/min to 130 °C (3 min hold time), and 10 °C/min to 220 °C (1 min hold time). The triple quadrupole was operated in both full scan acquiring in the range of *m*/*z* 30–300. The temperature of the transfer line, ion source, and quadrupoles were set at 260 °C, 250 °C, and 150 °C, respectively. Helium was used as the quench gas at a flow of 4 mL/min and nitrogen as the collision gas at a flow of 1.5 mL/min. HiSorb probes were conditioned under a N_2_ flow rate (70 mL/min) at 280 °C for 2 h.

### 4.5. Data Processing and Analysis

For NMR data, the identification of polymer signals was performed based on the literature [19,20,21,22,23,39,40]. The identification of minor compounds was supported by the available spectra on free NMR database (accessed on 1 December 2022. (https://sdbs.db.aist.go.jp/sdbs/cgi-bin/cre_index.cgi and https://hmdb.ca/spectra/nmr_one_d_search/new) and by using a ^1^H NMR spectra prediction tool (https://www.nmrdb.org/new_predictor/index.shtml?v=v2.138.0).

For HiSorb-TD-GC-MS data, MassHunter software (v. B.07.00, Agilent Technologies, Santa Clara, CA, USA) was used to acquire chromatographic data and perform qualitative analysis through the Agilent Unknown^TM^ tool. A deconvolution process was applied to computationally separate co-eluting compounds and create pure MS spectra. GC peaks with a signal-to-noise ratio higher than 10 and characterized by a NIST Mass Spectral Library (v.14) (Gaithersburg, MD, USA) match factor of at least 80% as well as a difference between the experimental and library Kovats retention index below 50 were selected for the subsequent analysis. The resulting data matrix was analyzed by R software (RStudio, Boston, MA, USA).

## Figures and Tables

**Figure 1 molecules-28-01382-f001:**
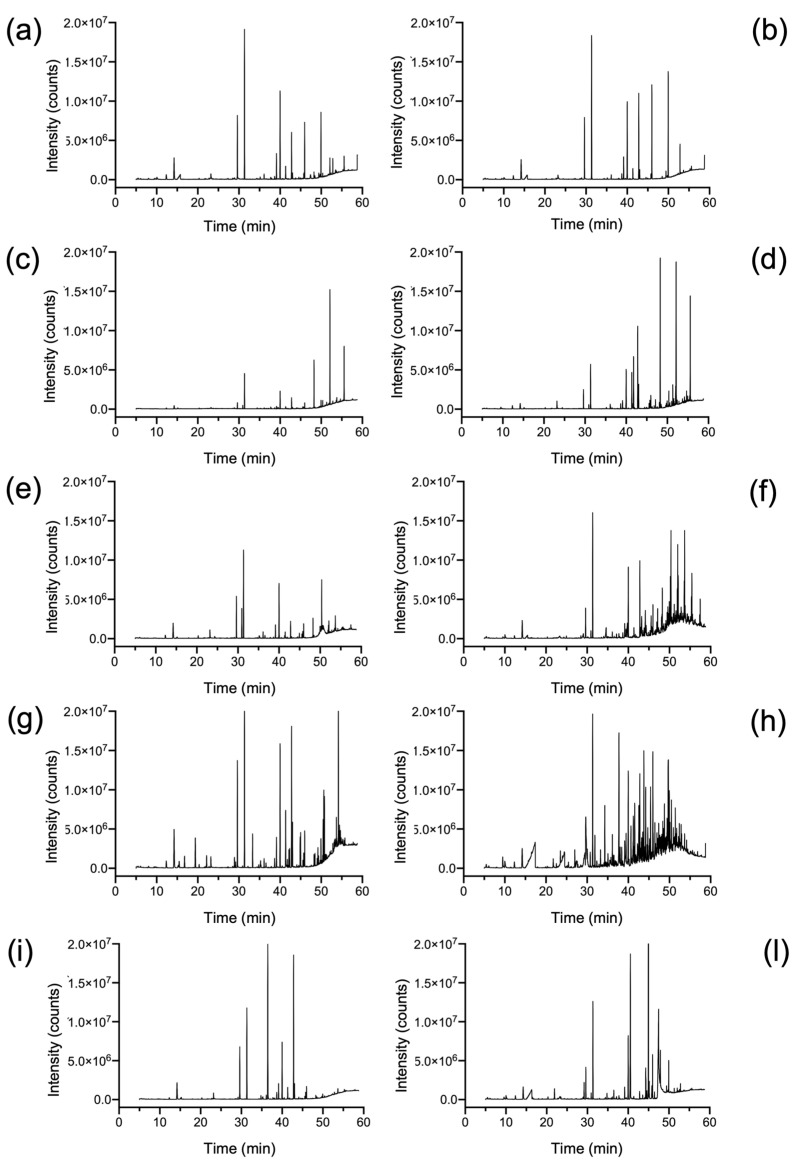
Total ion chromatograms of (**a**) PET (t_0_); (**b**) PET (t_0_ + 4 w); (**c**) HDPE (t_0_); (**d**) HDPE (t_0_ + 4 w); (**e**) LDPE (t_0_); (**f**) LDPE (t_0_ + 4 w); (**g**) PP (t_0_); (**h**) PP (t_0_ + 4 w); (**i**) PS (t_0_); and (**l**) PS (t_0_ + 4 w).

**Figure 2 molecules-28-01382-f002:**
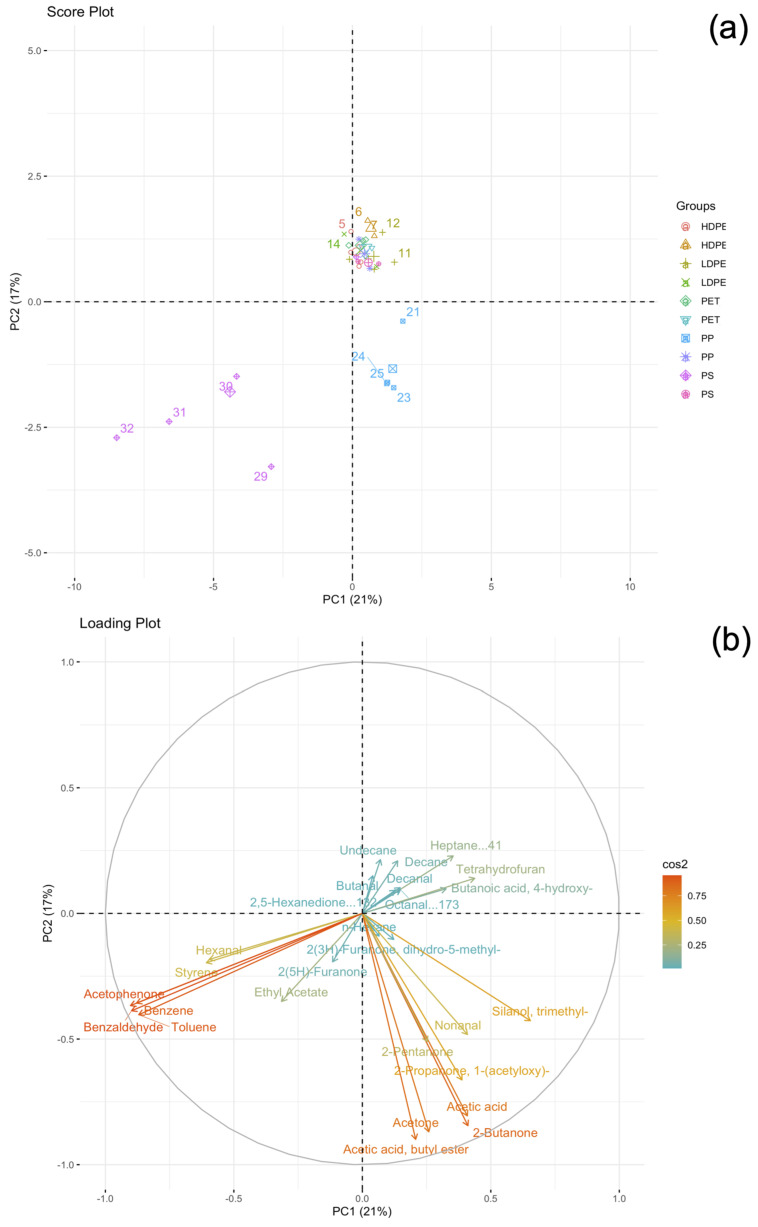
Principal component analysis of HiSorb-GC-MS data: (**a**) score plot and (**b**) loading plot.

**Figure 3 molecules-28-01382-f003:**
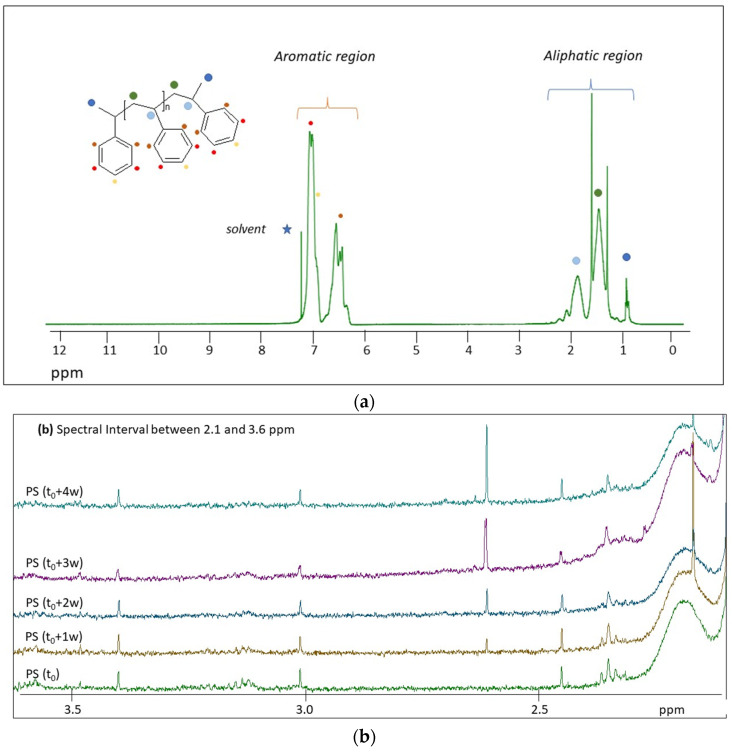
^1^H NMR spectra of PS microplastics during degradation. (**a**) ^1^H NMR spectrum of the pristine PS microplastics. Stackplot of the ^1^H NMR spectra of the PS microplastics at different times of irradiation (from bottom to top): t_0_, t_0_ + 1 w, t_0_ + 2 w, t_0_ + 3 w, and t_0_ + 4 w, in the spectral regions between 2.1 and 3.6 ppm (**b**), between 3.5 and 5 ppm (**c**), and between 7.0 and 8.5 ppm (**d**). Spectra were normalized with respect to the aromatic signal centered at 7.1 ppm. Stars signals are referred to solvent used for the ^1^H NMR analysis.

**Figure 4 molecules-28-01382-f004:**
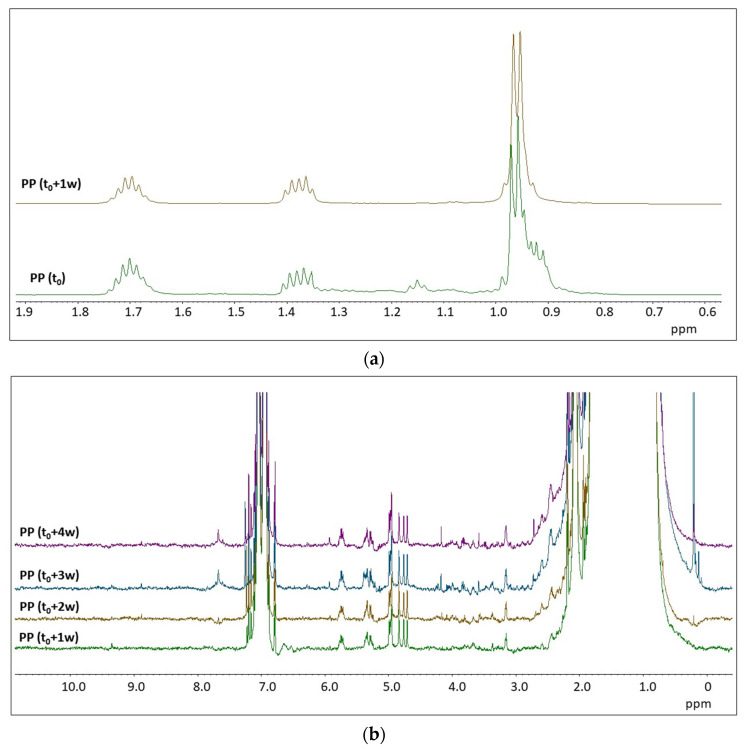
(**a**) ^1^H NMR spectra of the pristine PP microplastics (t_0_) and after 1 week of irradiation (t_0_ + 1 w). (**b**) Stackplot of the ^1^H NMR spectra of the PP microplastics at different times of irradiation (from bottom to top): t_0_ + 1 w, t_0_ + 2 w, t_0_ + 3 w, and t_0_ + 4 w, enlarged at the spectral baseline to see the presence of small signals. Spectra were normalized with respect to the aliphatic signal between 0.85 and 1.0 ppm (due to the CH_3_ in the PP). Note that the signals around 7 ppm and 2.3 ppm are due to the deuterated solvent (toluene-d_8_).

**Figure 5 molecules-28-01382-f005:**
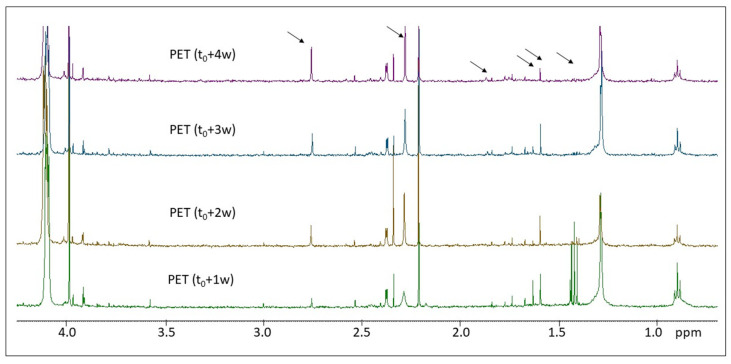
Stackplot of the ^1^H NMR spectra of the PET microplastics at different times of irradiation (from bottom to top): t_0_ + 1 w, t_0_ + 2 w, t_0_ + 3 w, and t_0_ + 4 w, enlarged at the spectral baseline in the region between 0.6 and 4.3 ppm. Spectra were normalized with respect to the aromatic signal centered at 8.11 ppm due to the aromatic moiety of PET. Dark arrows indicate the most significant signals appearing or disappearing as a result of degradation.

**Figure 6 molecules-28-01382-f006:**
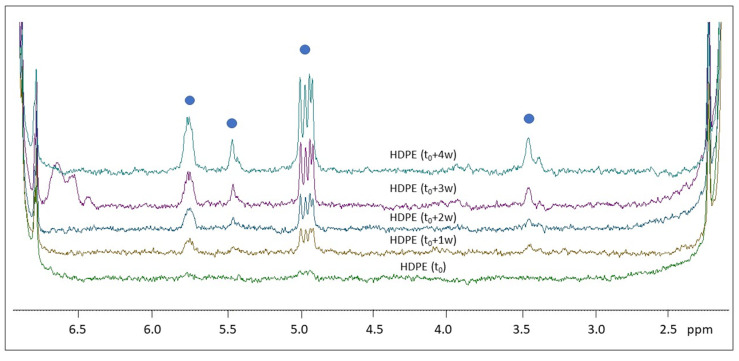
Stackplot of the ^1^H NMR spectra of the HDPE microplastics at different times of irradiation (from bottom to top): t_0_, t_0_ + 1 w, t_0_ + 2 w, t_0_ + 3 w, and t_0_ + 4 w, enlarged at the spectral baseline in the region between 2.0 and 7.0 ppm. Spectra were normalized with respect to the aliphatic signal centered at 1.3 ppm, due to the CH_2_ moiety of the HDPE chain. Blue circles indicate the most significant signals appearing as a result of degradation, as discussed in Section 3.

**Figure 7 molecules-28-01382-f007:**
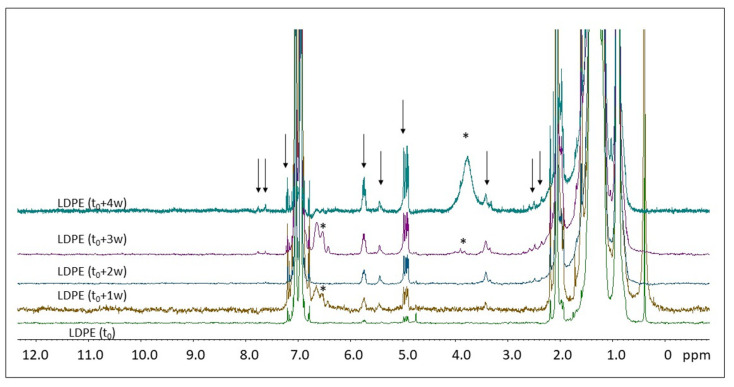
Stackplot of the ^1^H NMR spectra of the LDPE microplastics at different times of irradiation (from bottom to top): t_0_, t_0_ + 1 w, t_0_ + 2 w, t_0_ + 3 w, and t_0_ + 4 w, enlarged at the spectral baseline in the whole proton spectral region. Spectra were normalized with respect to the aliphatic signal centered at 1.3 ppm, due to the CH_2_ moiety of the LDPE chain. Dark arrows indicate the most significant signals appearing as a result of degradation, while asterisks (*) show anomalous signals in some of the analyzed microplastics.

**Figure 8 molecules-28-01382-f008:**
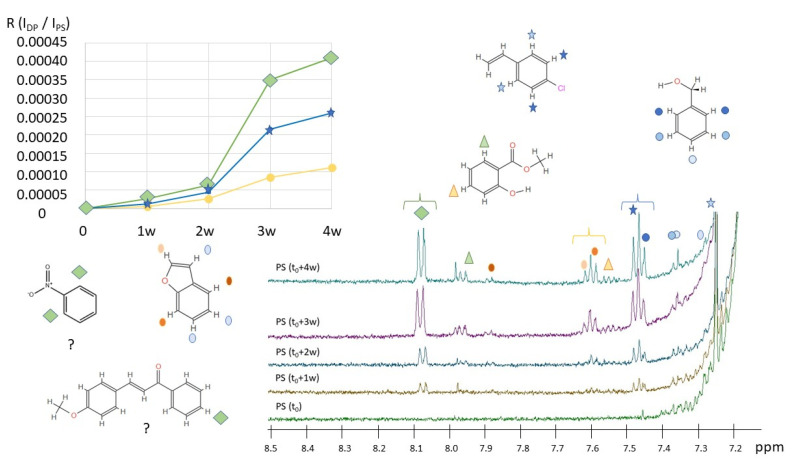
Enlargement of the ^1^H NMR spectra of pristine PS (t_0_) and PS after 1 w, 2 w, 3 w, and 4 w of degradation, from 7.2 to 8.5 ppm. Main ^1^H NMR aromatic signals have been assigned to compounds, whose molecular structure is shown. Hypothetical compounds are also reported as described in the text. The correspondence between each signal and the proton/protons on the molecular sketches is put in evidence through colored symbols. On the top left, the ratio R between the integral of the aromatic signals of PS (I_PS_) and the integral of the signals of degradation products (I_DP_), during the degradation time, from 0 (t_0_) to 4 weeks (t_0_ + 4 w) for three more significant signals, as indicated by the colored braces on top of the ^1^H NMR spectra.

**Figure 9 molecules-28-01382-f009:**
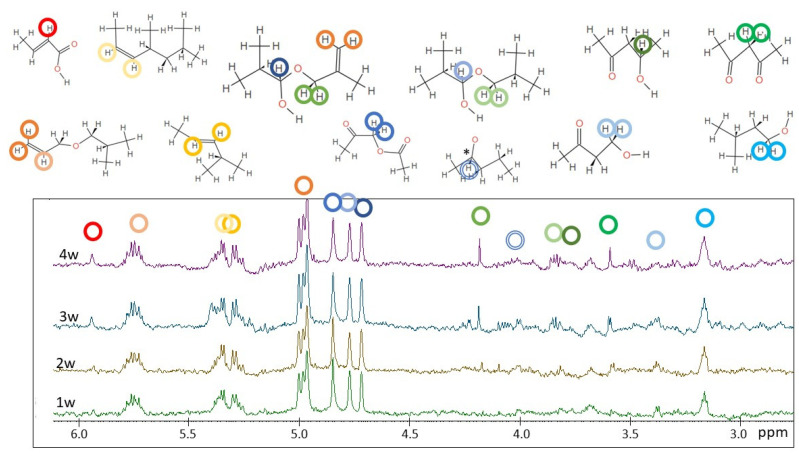
Enlargement of the ^1^H NMR spectra of PP after 1 w, 2 w, 3 w, and 4 w of irradiation, from 2.8 to 6.1 ppm. The main ^1^H NMR signals and the corresponding assignments to compounds with the given molecular structure are shown. The correspondence between each signal and the proton/protons on the molecular sketches is put in evidence through colored circles. See the text for further explanations.

**Figure 10 molecules-28-01382-f010:**
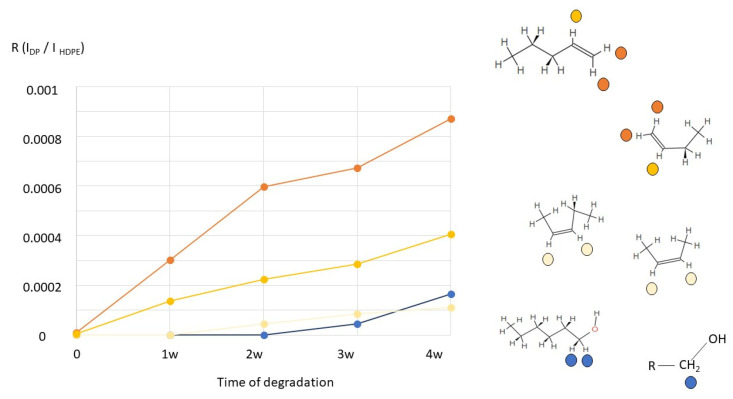
On the left, trends of the ratio R between the integral of the CH_2_- signals of HDPE (I_HDPE_) and the integral of the signals of degradation products (I_DP_), during the degradation time, from 0 (t_0_) to 4 weeks (t_0_ + 4 w). Four most significant signals have been selected, ascribable to alkene and alcohol products of degradation. On the right, molecular structure of representative molecules and proton signals under observation, as identified from ^1^H NMR.

**Figure 11 molecules-28-01382-f011:**
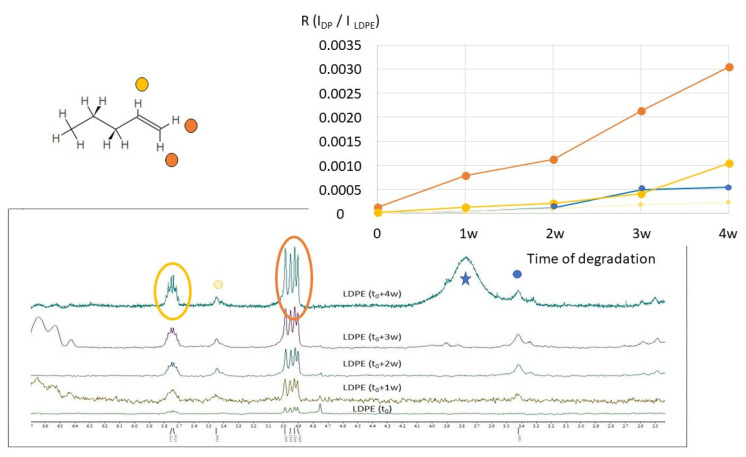
^1^H NMR spectra of the LDPE in the region between 2.4 ppm and 6.7 ppm. Signals ascribable to 1-alkene (such as 1-pentene, see the molecular sketch on the top) are marked with colored circles. Trends of the ratio R between the integral of the CH_2_- signals of LDPE (I_LDPE_) and the integral of the signals of degradation products (I_DP_), during the degradation time, from 0 (t_0_) to 4 weeks (t_0_ + 4 w), are reported on the right. Four most significant signals have been selected, ascribable to alkene and alcohol compounds. The large signals in the spectrum at t_0_ + 4 w (marked with a star) are due to water.

**Table 1 molecules-28-01382-t001:** Pristine plastic materials used in the present study, density, and average dimension of the plastic samples.

Plastic Material	Density (g/cm^3^)	Average Dimension of Plastic Samples (μm)
Low density polyethylene (LDPE)	0.917	632
High density polyethylene (HDPE)	0.952	622
Polyethylene terephthalate (PET) (About 5% crystallinity as provided)	1.28	509
Polypropylene (PP)	0.900	857
Polystyrene (PS)	1.05	564

## Data Availability

Not available.

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
