# Peer review of "Multi-Analytical Approach to Characterize the Degradation of Different Types of Microplastics: Identification and Quantification of Released Organic Compounds"

_molecules, 2023, doi:10.3390/molecules28031382_

Round 1

Reviewer 1 Report

The paper manuscript “Multi-analytical approach to characterize the degradation of 2different types of microplastics: identification and quantification of released organic compounds.”

Overall, this is a well written manuscript and has a potential to be accepted. 

Nevertheless, the authors should revise better their Abstract, Introduction and Discussion. The study must summarize and clearly present the findings (regarding the degradation products) related to each method used. Furthermore, the study must examine how the findings relate to previous research in this area. 

In addition, several comments follow.

1) General Comment: Please check abbreviations with consistency in main text. Define it at the first appearance, then use it after the definition (e.g. MPs, NPs, HDPE, PE, PET, PP, PS, LDPE, Hisorb-TD-GC-MS, HS, t0, 4w, PC1, PC2, NTME, etc.).

2) Lines 131-134 or/and Introduction: Please provide further data regarding the NTME technique used in a former study. 

3) Lines 193, 224, 245, 265: A space should be added.

4) Lines 478-481: Please provide an appropriate reference, if available.

5) Line 539: The abbreviation (SIM), is proposed to be added. 

6) Further data regarding the selected SIM transitions of each analyte tested could be added in the Supplementary. 

7) You should define if the methods are quantitative.  If they are, further data regarding findings (declaring the concentrations of the degradation products found) may be added in the Supplementary with the relevant validation data. For example, Table S1 in the Supplementary may be amended with additional data. 

8) Lines 545-549: Please provide the proper references.

9) Supplementary, Table S1: Abbreviations of the first row must be defined, where needed. Also a typo error “ ….at t0 e after …”, must be corrected. 

Author Response

Reviewer #1:

The paper manuscript “Multi-analytical approach to characterize the degradation of different types of microplastics: identification and quantification of released organic compounds.”

Overall, this is a well written manuscript and has a potential to be accepted.

Nevertheless, the authors should revise better their Abstract, Introduction and Discussion. The study must summarize and clearly present the findings (regarding the degradation products) related to each method used. Furthermore, the study must examine how the findings relate to previous research in this area.

Authors: The manuscript has been revised in view of the present comment. In particular, we have added a paragraph at the end of the Discussion section underlining the main finding and novelty of the present research work. A paragraph is also added to put in evidence the improvements of this work with respect to the actual literature.

In addition, several comments follow.

1) General Comment: Please check abbreviations with consistency in main text. Define it at the first appearance, then use it after the definition (e.g. MPs, NPs, HDPE, PE, PET, PP, PS, LDPE, Hisorb-TD-GC-MS, HS, t0, 4w, PC1, PC2, NTME, etc.).

Authors: done.

2) Lines 131-134 or/and Introduction: Please provide further data regarding the NTME technique used in a former study.

Authors: done.

3) Lines 193, 224, 245, 265: A space should be added.

Authors: done.

4) Lines 478-481: Please provide an appropriate reference, if available.

Authors: done.

5) Line 539: The abbreviation (SIM), is proposed to be added.

Authors: done.

6) Further data regarding the selected SIM transitions of each analyte tested could be added in the Supplementary.

Authors: GC-MS chromatograms were acquired in both full scan and SIM mode, which can be used to monitor specific VOCs potentially of interest for the analysis of head-space samples collected from plastic materials. Nevertheless, in this manuscript we mainly focused on the preliminary qualitative chemical characterization of VOCs profile released from certain types of microplastics during the artificial degradation procedure. Further experiments will be focused to improve the initial SIM method including some of the tentatively identified compounds to determine their concentration. For these reasons we did not include in the submitted version of the manuscript any of SIM transitions. Anyway, all the MS parameters are discussed in the article number 42.

7) You should define if the methods are quantitative. If they are, further data regarding findings (declaring the concentrations of the degradation products found) may be added in the Supplementary with the relevant validation data. For example, Table S1 in the Supplementary may be amended with additional data.

Authors: our main idea was to evaluate the potential use of two complementary instrumental techniques, i.e., GC-MS and 1H NMR, to better characterize the degradation products of MPs. Next steps will be surely focused to use both protocols for the determination of degradation compounds over-time providing additional data on the MPs degradation processes. In particular, the next step of the research will focus on the quantification of some of the identified compounds produced during degradation by means of both techniques, in order to better understand the mechanisms of degradation.

8) Lines 545-549: Please provide the proper references.

Authors: References were included at the end of the sentence.

9) Supplementary, Table S1: Abbreviations of the first row must be defined, where needed. Also a typo error “ ….at t0 e after …”, must be corrected.

Authors: done. Thanks to this comment, we reviewed the table caption.

Reviewer 2 Report

The researcher Giulia Giaganini and co-workers have a report entitled “Multi-analytical approach to characterize the degradation of different types of microplastics: identification and quantification of released organic compounds”. This paper extensively studies the quantification and identification of the degradation of microplastics by temperature and light. The researchers took a series of reference polymers such as HDPE, LDPE, polypropylene, Polystyrene, and polyethylene terephthalates.  The time-dependent degradation was monitored (characterized) with multi-analytical technics, particularly by 1HNMR spectroscopy and TD-GCMS analytical methods.

The research article is well-presented and clearly written with appropriate literature and experimental characterization, and analysis data. The NMR and GC-MS data were well interpreted and discussed in appropriate places wherever required for the discussion of the supportive claims (identification) of degradation. Therefore, this work has a high potential to be publishable in MDPI’s molecules, so I accept the present article for publication.   

Author Response

Reviewer #2:

The researcher Giulia Giaganini and co-workers have a report entitled “Multi-analytical approach to characterize the degradation of different types of microplastics: identification and quantification of released organic compounds”. This paper extensively studies the quantification and identification of the degradation of microplastics by temperature and light. The researchers took a series of reference polymers such as HDPE, LDPE, polypropylene, Polystyrene, and polyethylene terephthalates.  The time-dependent degradation was monitored (characterized) with multi-analytical technics, particularly by 1HNMR spectroscopy and TD-GCMS analytical methods.

The research article is well-presented and clearly written with appropriate literature and experimental characterization, and analysis data. The NMR and GC-MS data were well interpreted and discussed in appropriate places wherever required for the discussion of the supportive claims (identification) of degradation. Therefore, this work has a high potential to be publishable in MDPI’s molecules, so I accept the present article for publication.  

Authors: we would like to thank the reviewer for appreciating the quality of our manuscript.

Reviewer 3 Report

The results are interesting and some reasonable explanation is provided. It is acceptable with revision. However, several modifications are required as follows:

1.      Pictures need to be clearer and more attractive, to facilitate reading.

2.      Are there any released organic compounds during degradation, which have been mineralized into CO2 and H2O?

3.      More significant works should be introduced to keep abreast of the latest research trends. e.g.: Chinese Journal of Catalysis, 2022, 43, 2652–2664; Chemical Engineering Journal, 452 (2023) 139417.

4.      The advantages and disadvantages of the analysis method should be discussed.

5.      The manuscript has some grammatical and linguistic errors and needs to be checked and revised thoroughly.

6.      Are there any hazardous organics produced during degradation?

7.      Why not investigate the plastic degradation under natural conditions?

8.      More published works should be listed for comparing with this work.

Author Response

Reviewer #3:

The results are interesting and some reasonable explanation is provided. It is acceptable with revision. However, several modifications are required as follows:

  1. Pictures need to be clearer and more attractive, to facilitate reading.

Authors: we tried our best to prepare the figures aimed at highlighting the main findings of our study and at the same time facilitating reading. Thus, any suggestion from the reviewer is welcomed, we will change figures accordingly.

  1. Are there any released organic compounds during degradation, which have been mineralized into CO2 and H2O?

Authors: no, we did not mineralize any degradation compounds into CO2 and H2O.

  1. More significant works should be introduced to keep abreast of the latest research trends. e.g.: Chinese Journal of Catalysis, 2022, 43, 2652–2664; Chemical Engineering Journal, 452 (2023) 139417.

Authors: we would like to thank the reviewer for sharing with us two interesting papers. We believe that the suggested articles did not fit properly the main scope of the proposed manuscript, thus we did not include them in the reference list. Anyway, if the editor thinks differently, we will update the reference list in the revised version of the manuscript as suggested.

  1. The advantages and disadvantages of the analysis method should be discussed.

Authors: done. We added a small paragraph explaining the main findings of our study and the advantage of our approach.

  1. The manuscript has some grammatical and linguistic errors and needs to be checked and revised thoroughly.

Authors: done. We checked and revised the grammatical and linguistic errors.

  1. Are there any hazardous organics produced during degradation?

Authors, yes. Some of the compounds released during the degradation process can be harmful and toxic.

  1. Why not investigate the plastic degradation under natural conditions?

Authors, the degradation conditions investigated in our study were selected to simulate natural aging occurring for marine plastic debris. Anyway, potential deviations from natural aging conditions cannot be excluded since we used a confined environment in which the irradiation could be continue and intense.

  1. More published works should be listed for comparing with this work.

Authors, we already cited most of the recent articles published in this field. We did not include any additional articles to avoid the increasing of the reference list. We believe that 42 articles are enough for a complete understanding of the topic.

Round 2

Reviewer 1 Report

The paper manuscript “Multi-analytical approach to characterize the degradation of different types of microplastics: identification and quantification of released organic compounds.”

Overall, this is a well written manuscript and is considered  to be accepted with minor revision.

However, following author's clarification that  in this manuscript you mainly focused on the preliminary qualitative chemical characterization of VOCs profile released from certain types of microplastics during the artificial degradation procedure, reference to SIM mode (if not used for VOCs profile) is proposed to be removed  to avoid misuderstunting (see point 4.4). It should be clearly stated which analytical method have been used in this manuscript  in relation with what had been used in the reference 42 and what is proposed to be done in the future. 

Author Response

Reviewer #1:
The paper manuscript “Multi-analytical approach to characterize the degradation of different types of microplastics: identification and quantification of released organic compounds.”

Overall, this is a well written manuscript and is considered to be accepted with minor revision.

However, following author's clarification that in this manuscript you mainly focused on the preliminary qualitative chemical characterization of VOCs profile released from certain types of microplastics during the artificial degradation procedure, reference to SIM mode (if not used for VOCs profile) is proposed to be removed to avoid misuderstunting (see point 4.4). It should be clearly stated which analytical method have been used in this manuscript in relation with what had been used in the reference 42 and what is proposed to be done in the future.

Authors: done. We removed the SIM mode as suggested by the reviewer. We modified paragraph 4.4 to better explain that we performed a preliminary screening of VOCs released during degradation. Moreover, we clearly stated that the GC-MS method was already reported in ref 42.

Reviewer 3 Report

Although much work has been done, I think this work needs improvement.

1. Please explain why the authors investigates the microplastics degradation at 40 °C, which is not the natural temperature. To study the microplastics degradation process at natural temperature is more reasonable.

2. More significant works should be introduced to keep abreast of the latest research trends. e.g.: Chinese Journal of Catalysis, 2022, 43, 2652–2664; Chemical Engineering Journal, 2023,452,139417,Molecules,2023, 28(1), 239

3. The toxicity variations before aging and after aging produced during degradation?

Author Response

Reviewer #3:
Although much work has been done, I think this work needs improvement.

1. Please explain why the authors investigates the microplastics degradation at 40 °C, which is not the natural temperature. To study the microplastics degradation process at natural temperature is more reasonable.

Authors: we would like to thank the reviewer for the comment. We investigated 40 °C because it is a reasonable temperature of the sand during the summer period. Moreover, please consider that plastic debris may also increase circadian temperature in beach sediments as recently discussed by Lavers et al (https://doi.org/10.1016/j.jhazmat.2021.126140).

2. More significant works should be introduced to keep abreast of the latest research trends. e.g.: Chinese Journal of Catalysis, 2022, 43, 2652–2664; Chemical Engineering Journal, 2023,452,139417,Molecules,2023, 28(1), 239

Authors: done.

3. The toxicity variations before aging and after aging produced during degradation?

Authors: the main purpose of the manuscript was to perform a screening of the degradation products with HiSorb-TD-GC-MS and 1H NMR. Our data highlighted the release of harmful compounds as reported for PS-based material that released mainly aromatic compounds such as acetophenone, benzaldehyde, toluene, and benzene. So, it is possible to speculate that degradation processes lead to an increase of the intrinsic toxicity of the plastic debris.